# Superior cardiovascular protection with GLP-1 RAs over SGLT2 inhibitors in DM and HFpEF: A propensity score matching study

Allen Cheng-Wei Li[1,2☯], Yang-Chi Lin[1☯], Jing-Yang Huang [3,4], Lung-Ching Chen[5,6], Su-Kiat Chua [5,6]*

1 School of Medicine, Chung Shan Medical University, Taichung, Taiwan, 2 Department of General Medicine, Shin Kong Wu Ho Su Memorial Hospital, Taipei, Taiwan, 3 Institute of Medicine, Chung Shan Medical University, Taichung, Taiwan, 4 Center for Health Data Science, Department of Medical Research, Chung Shan Medical University Hospital, Taichung, Taiwan, 5 School of Medicine, College of Medicine, Fu Jen Catholic University, New Taipei, Taiwan, 6 Division of Cardiology, Department of Internal Medicine, Shin Kong Wu Ho-Su Memorial Hospital, Taipei, Taiwan

☯ These authors contributed equally to this work.
* benchua1131@hotmail.com

## Abstract

### Background

Heart failure with preserved ejection fraction (HFpEF) and diabetes mellitus (DM) are interrelated conditions associated with high morbidity and mortality. This study compared the cardiovascular protective effects of glucagon-like peptide-1 receptor agonists (GLP-1 RAs) versus sodium-glucose cotransporter-2 (SGLT2) inhibitors in this population.

### Methods

This retrospective cohort study used data from the TriNetX database. It included 2,177 matched pairs of patients with HFpEF and DM treated with either GLP-1 RAs or SGLT2 inhibitors. Outcomes assessed over three years were a composite of all-cause mortality and progression to systolic heart failure, acute myocardial infarction, or stroke.

### Results

GLP-1 RAs significantly reduced the risk of composite outcomes at one year (Hazard Ratio, HR 0.784; 95% CI, 0.658–0.934), two years (HR 0.813; 95% CI, 0.702–0.941), and three years (HR 0.825; 95% CI, 0.717–0.950). Specifically, GLP-1 RAs showed significantly reduced risks of progression to systolic heart failure (HR 0.60) and stroke (HR 0.75) compared to SGLT2 inhibitors. These protective effects were most pronounced in the first year and showed a slightly diminishing trend. While not statistically significant, GLP-1 RAs also exhibited a trend towards fewer myocardial

**Data availability statement:** All relevant data are within the paper and its Supporting information files.

**Funding:** The author(s) received no specific funding for this work.

**Competing interests:** The authors have declared that no competing interests exist.

**Abbreviations:** HFpEF, Heart failure with preserved ejection fraction; DM, diabetes mellitus; GLP-1 RAs, glucagon-like peptide-1 receptor agonists; SGLT2, Sodium-glucose cotransporter-2; HFmrEF, Mildly reduced ejection fraction (HFmrEF); HFpEF, Heart failure with preserved ejection fraction; HCOs, Global healthcare organizations; MI, Myocardial Infarction; CKD, chronic kidney disease.

infarctions (HR 0.83) and lower mortality rates (HR 0.83) than SGLT2 inhibitors. Subgroup analyses revealed more significant benefits in patients aged ≥60, women, Caucasians, those without moderate-to-severe chronic kidney disease or chronic ischemic heart disease, and those with better-controlled DM.

## Conclusions

Among HFpEF patients with DM, GLP-1 RAs demonstrated superior cardiovascular protective effects compared with SGLT2 inhibitors over a 3-year follow-up period. Further randomized trials are required to confirm these findings.

---

## Introduction

The spectrum of heart failure encompasses distinct categories, such as heart failure with a reduced ejection fraction (HFrEF), heart failure with mildly reduced ejection fraction (HFmrEF), and heart failure with preserved ejection fraction (HFpEF). HFpEF is more prevalent in older adults, women, and obese individuals and is often associated with various comorbidities, including diabetes mellitus (DM). DM affects nearly 25% of patients diagnosed with HFpEF, contributing to worse outcomes due to shared pathophysiological mechanisms [1]. HFpEF is commonly associated with cardiovascular comorbidities such as atrial fibrillation, coronary artery disease, and chronic kidney disease, as well as non-cardiovascular conditions, including thyroid dysfunction and chronic obstructive pulmonary disease [2]. HFpEF and DM share common pathophysiological mechanisms, including systemic inflammation, myocardial fibrosis, endothelial dysfunction, and insulin resistance [3]. These processes contribute to myocardial stiffness, impaired relaxation, and microvascular dysfunction, which exacerbate heart failure symptoms and increase cardiovascular risks [4]. While SGLT2 inhibitors have demonstrated benefits in HFpEF by improving volume regulation and reducing hospitalization rates, Glucagon-Like Peptide-1 receptor agonists (GLP-1 RAs) offer cardiometabolic advantages, including anti-inflammatory and anti-atherosclerotic effects, improved insulin sensitivity, and weight loss [5]. Given these distinct mechanisms, directly comparing these therapies is crucial to identifying the optimal strategy for improving cardiovascular outcomes in patients with HFpEF and DM.

To date, no effective treatment specifically targeting HFpEF has been established [6–8]. Consequently, the current clinical emphasis is on symptom alleviation, functional enhancement, and minimizing HF-related hospitalizations. The EMPEROR-Preserved trial demonstrated a significant 29% reduction in HF-related hospitalizations with empagliflozin, highlighting the efficacy of SGLT2 inhibitors in improving clinical outcomes in HFpEF patients, regardless of diabetes status [5]. In 2023, the STEP-HFpEF trial showed that semaglutide, a once-weekly long-acting GLP-1 RAs, significantly improved exercise capacity, enhanced quality of life, and notably reduced body weight while mitigating inflammation over 52 weeks in obese patients with HFpEF compared to placebo [9]. This finding underscores the critical

need to evaluate whether alternative therapies, such as GLP-1 RAs, could offer comparable or superior benefits. By directly comparing these two therapeutic classes, our study aims to fill this knowledge gap and provide evidence for personalized treatment strategies in HFpEF patients with diabetes.

Direct head-to-head randomized controlled trials (RCTs) comparing GLP-1 RAs and SGLT2 inhibitors in HFpEF patients are limited, posing challenges in determining the optimal treatment strategy. Our study addresses this gap by directly comparing the long-term cardiovascular outcomes of these two therapies, providing insights to guide clinical decision-making and improve outcomes in this high-risk population.

## Materials and methods

### Data collection

The TriNetX database, which includes medical records from global healthcare organizations (HCOs) covering over 250 million patients, was used in this retrospective cohort study. The de-identified data, compliant with Section 164.514(a) of the Health Insurance Portability and Accountability Act (HIPAA) Privacy Rule, imply that Institutional Review Board approval and participant consent were not required for this study. Data were collected on a specified date from the US Collaborative Network, encompassing over 112 million patients from 61 HCOs, covering the period from March 1, 2013, to December 31, 2022. Data extraction was performed on December 20, 2023, using codes from several coding classification systems, including the International Classification of Diseases, Tenth Revision, Clinical Modification (ICD-10-CM), Veterans Affairs (VA) Drug Classification System, and Anatomical Therapeutic Chemical (ATC) Classification System.

### Population and study design

In this study, we included patients aged 20 years or older with at least one documented instance of diastolic heart failure and type 2 diabetes. Long-term use was defined as having at least one prescription issued six months to one year after the initial prescription. We excluded individuals with systolic heart failure (SHF) (ICD-10-CM: I50.3), acute myocardial infarction (MI) (I21), other acute ischemic heart diseases (I24), or stroke (I61-I63) before their first recorded prescription of GLP-1 RAs and SGLT2 inhibitors, which was designated as the index date for both cohorts. Additionally, from March 1, 2013, to December 31, 2022, patients prescribed SGLT2 inhibitors in the GLP-1 RAs cohort and vice versa were excluded. Flowcharts illustrating the establishment of both cohorts are presented in Fig 1. The figure outlines the cohort selection process, detailing the inclusion and exclusion criteria applied to the study population. It also highlights the final matched cohorts used for analysis, ensuring a clear understanding of the steps in forming the study groups.

### Propensity-score matching

The GLP-1 RAs and SGLT2 inhibitor cohorts were meticulously 1:1 propensity-matched, incorporating three years of records before the index date. The matching process accounted for demographics (age and racial categories, including White, Black, African American, and Asian), diagnoses, and laboratory data. Particular attention was given to confounders related to HF progression, including conditions such as primary hypertension, hyperlipidemia, acute myocardial infarction, chronic ischemic heart disease, atrial fibrillation and flutter, chronic kidney disease with staging, and nonrheumatic aortic and mitral valve disorders. This rigorous approach ensured balanced cohorts for comparative analysis. Numerical laboratory data, such as systolic and diastolic blood pressure, body mass index (BMI), and glycated hemoglobin (HbA1c), were included as critical cardiovascular and metabolic health markers. Elevated blood pressure is associated with increased left ventricular hypertrophy and stiffness, BMI reflects obesity-related cardiac stress, and HbA1c indicates glycemic control, which are key factors influencing HF progression. The specific codes applied from the various classification systems are detailed in the Supplementary Data (Appendix 1 in S1 File).

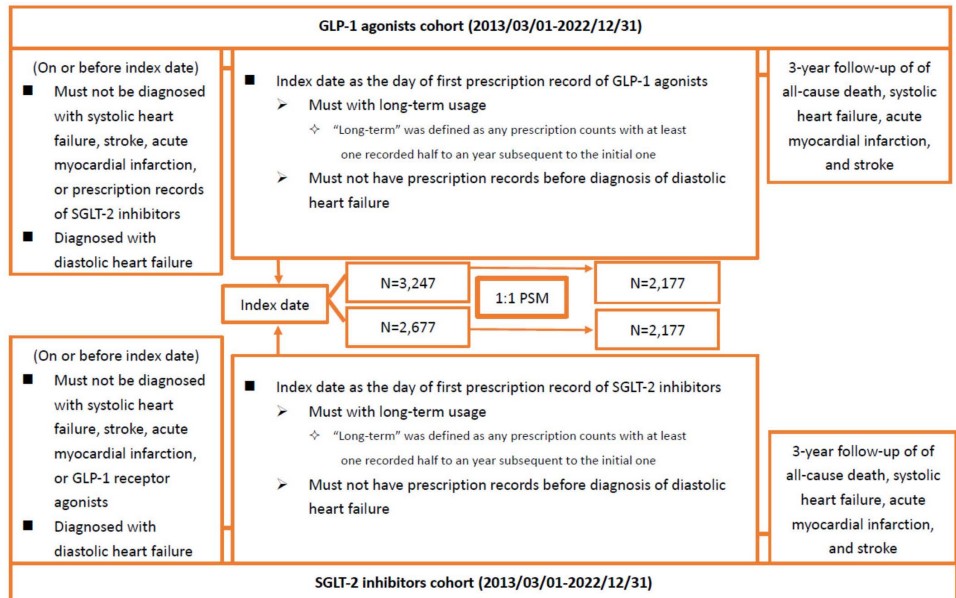

**Fig 1. Flowchart of Cohort Construction.** This figure depicts the process of cohort construction, detailing the inclusion and exclusion criteria applied to build the study cohorts for patients treated with GLP-1 receptor agonists (GLP-1 RAs) and SGLT2 inhibitors. GLP-1 RAs, glucagon-like peptide-1 receptor agonists; SGLT2 inhibitors, sodium-glucose cotransporter-2 inhibitors.

## Outcomes measurement

This retrospective cohort study evaluated the risk of all-cause mortality, progression to SHF, acute MI, and stroke in patients with DM and DHF following their first prescription of GLP-1 RAs or SGLT2 inhibitors. The primary outcome was a composite measure of these events, while secondary outcomes assessed each event individually. Outcome measurement spanned three years from the index date, defined as the day of the first prescription. Detailed codes for outcome definitions are provided in Supplementary Data (Appendix 1 in S1 File).

## Subgroup analysis

Subgroup analyses of primary outcomes were conducted to assess variations in findings based on sex, race (white vs. non-white), age (≥60 years), blood sugar control (HbA1c < 8.0% vs. ≥ 8.0%), renal function (moderate-to-severe chronic renal disease), and chronic ischemic heart disease. These analyses were performed after 1:1 propensity score matching to ensure balance.

## Statistical analysis

The comparability of baseline characteristics between the two matched cohorts was evaluated using the standardized mean difference (SMD), with an SMD below 0.1 indicating balance. Kaplan-Meier analyses assessed the five-year risk of outcomes, presented as survival probabilities with time-varying hazard ratios (HRs) and 95% confidence intervals using a Cox proportional hazard model. The statistical significance of survival curves was determined via log-rank tests with a p-value threshold of 0.05. All analyses were conducted using the TriNetX analytics platform, which facilitates data extraction, cohort matching, and statistical evaluation. The figures were generated using R Studio (R version 4.2.2) based on data exported from TriNetX. This workflow ensured accurate analyses and reproducibility of the results while leveraging the visualization capabilities of R Studio for clear data presentation.

## Results

### Baseline characteristics

The US Collaborative Network from TriNetX included 3,247 patients in the GLP-1 RAs cohort and 2,677 in the SGLT2 inhibitor cohort. Most patients were in their sixties (63.4 vs. 67.6 years old) and predominantly White (60.6% vs. 60.8%). The GLP-1 RAs cohort had a higher proportion of females (56.9% vs. 47.9%). Primary hypertension was the most common comorbidity (91.1% vs. 91.8%), followed by dyslipidemia (85.7% vs. 84.9%), chronic ischemic heart disease (42.4% vs. 49.6%), and stage 3 CKD (31.2% vs. 30.2%). Mean HbA1c levels were 8.2 and 7.8, with mean BMIs of 38.6 and 35.2, respectively, although BMI data were available for only 36.9% and 41.4% of patients.

After propensity score matching, the GLP-1 RAs and SGLT2 inhibitor cohorts each contained 2,177 patients, with all characteristics and numeric laboratory data comparable, as evidenced by all SMDs being below 0.1. Table 1 shows the detailed characteristics before and after propensity score matching.

### Primary analysis

GLP-1 RAs demonstrated a protective effect on the primary outcome in diabetic patients with HFpEF compared to SGLT2 inhibitors (HR = 0.71). At the one-year post-index date, 201 patients in the GLP-1 RAs cohort and 334 in the SGLT2 inhibitor cohort experienced composite cardiovascular events. By the 2- and 3-year follow-ups, these numbers increased to 368 vs. 497 and 466 vs. 555, respectively, with the hazard ratio gradually increasing, suggesting a reduction in profit over time.

For secondary outcomes, GLP-1 RAs significantly reduced the risks of systolic heart failure (SHF, HR = 0.60) and stroke (HR = 0.75) compared to SGLT2 inhibitors. However, no significant differences were observed in all-cause mortality (HR = 0.83) or acute myocardial infarction (HR = 0.83). Kaplan-Meier analysis (Fig 3) confirmed these findings, with GLP-1 RAs significantly lowering the risk of primary composite cardiovascular events (P < 0.001), SHF (P < 0.001), and stroke (P = 0.035), while differences in mortality (P = 0.13) and acute MI (P = 0.10) remained non-significant during the 3-year follow-up.

In summary, GLP-1 RAs demonstrated superior cardiovascular benefits compared to SGLT2 inhibitors in reducing the risk of cardiovascular events, SHF, and strokes in diabetic patients with HFpEF, as shown in Figs 2 and 3.

### Subgroup analyses

Subgroup analyses revealed notable variations in patient demographics and conditions. In patients aged > 60 years, GLP-1 RAs were more effective in reducing the risk of composite cardiovascular events (HR = 0.66) than in younger patients (HR = 0.72). Sex-based differences were apparent, with female patients exhibiting slightly lower HRs (HR = 0.61) than male patients (HR = 0.72). Patients with white ethnic backgrounds presented a significant risk reduction (HR = 0.73), while others did not (black HR = 0.79, others HR = 0.80). Moreover, patients without preexisting conditions, such as moderate-to-severe CKD and chronic ischemic heart disease, showed more pronounced risk reductions (HR = 0.69 and HR = 0.66, respectively) than their counterparts (HR = 0.75 and HR = 0.80, respectively). Additionally, the benefits of GLP-1 RAs were less significant in patients with suboptimal DM control (HbA1c ≥ 8.0%) (HR = 0.80) compared with those of better control (HR = 0.73). Details of the subgroup analyses are shown in Fig 4.

## Discussion

Our results demonstrate that GLP-1 receptor agonists provide better cardiovascular protection than SGLT2 inhibitors in patients with HFpEF and DM, significantly reducing the risk of cardiovascular events, progression to SHF, and stroke. While the benefits were strongest in the initial treatment stages, there was a slight reduction in efficacy over the 3-year follow-up. Additionally, GLP-1 RAs showed a trend towards fewer MI and lower mortality rates, though these findings were not statistically significant.

**Table 1. Baseline Characteristics of Patients Treated with GLP-1 Receptor Agonists and SGLT2 Inhibitors Before and After Propensity Score Matching.**

| Variables | Before the propensity score match | | | After the propensity score match | | |
|---|---|---|---|---|---|---|
| | GLP-1 RAs (n = 3,247) | SGLT2 inhibitors (n = 2,677) | SMD | GLP-1 RAs (n = 2,177) | SGLT2 inhibitors (n = 2,177) | SMD |
| **Age at Index** | 63.4 +/- 10.8 | 67.6 +/- 10.8 | 0.389 | 65.7 +/- 10.1 | 65.8 +/- 10.7 | 0.008 |
| **Race, n (%)** | | | | | | |
| White | 1969(60.6) | 1627(60.8) | 0.003 | 1333(61.2) | 1321(60.7) | 0.011 |
| Black or African American | 750(23.1) | 592(22.1) | 0.024 | 494(22.7) | 499(22.9) | 0.005 |
| Asian | 49(1.5) | 86(3.2) | 0.112 | 46(2.1) | 40(1.8) | 0.02 |
| **Sex, n (%)** | | | | | | |
| Female | 1847(56.9) | 1283(47.9) | 0.18 | 1116(51.3) | 1123(51.6) | 0.006 |
| Male | 1302(40.1) | 1283(47.9) | 0.158 | 981(45.1) | 975(44.8) | 0.006 |
| **Comorbidities, n (%)** | | | | | | |
| Essential (primary) Hypertension | 2957(91.1) | 2457(91.8) | 0.025 | 1997(91.7) | 1992(91.5) | 0.008 |
| Disorders of lipoprotein metabolism and other lipidemias | 2783(85.7) | 2272(84.9) | 0.024 | 1852(85.1) | 1849(84.9) | 0.004 |
| Chronic ischemic heart disease | 1378(42.4) | 1329(49.6) | 0.145 | 993(45.6) | 1003(46.1) | 0.009 |
| Atrial fibrillation and flutter | 809(24.9) | 895(33.4) | 0.188 | 633(29.1) | 626(28.8) | 0.007 |
| Nonrheumatic mitral valve disorders | 433(13.3) | 486(18.2) | 0.133 | 335(15.4) | 316(14.5) | 0.024 |
| Nonrheumatic aortic valve disorders | 420(12.9) | 446(16.7) | 0.105 | 313(14.4) | 316(14.5) | 0.004 |
| Angina pectoris | 285(8.8) | 259(9.7) | 0.031 | 196(9) | 201(9.2) | 0.008 |
| Occlusion and stenosis of precerebral arteries, not resulting in cerebral infarction | 228(7) | 187(7) | 0.001 | 142(6.5) | 147(6.8) | 0.009 |
| Sequelae of cerebrovascular disease | 35(1.1) | 30(1.1) | 0.004 | 22(1) | 26(1.2) | 0.018 |
| Other cerebrovascular diseases | 104(3.2) | 97(3.6) | 0.023 | 76(3.5) | 70(3.2) | 0.015 |
| Chronic kidney disease, stage 1 | 34(1) | 47(1.8) | 0.06 | 30(1.4) | 31(1.4) | 0.004 |
| Chronic kidney disease, stage 2 | 242(7.5) | 215(8) | 0.022 | 162(7.4) | 172(7.9) | 0.017 |
| Chronic kidney disease, stage 3 | 1012(31.2) | 809(30.2) | 0.021 | 643(29.5) | 646(29.7) | 0.003 |
| Chronic kidney disease, stage 4 | 328(10.1) | 135(5) | 0.192 | 132(6.1) | 130(6) | 0.004 |
| Chronic kidney disease, stage 5 | 96(3) | 24(0.9) | 0.15 | 19(0.9) | 24(1.1) | 0.023 |
| End stage renal disease | 204(6.3) | 66(2.5) | 0.187 | 66(3) | 64(2.9) | 0.005 |
| **Hemoglobin A1c/Hemoglobin.total in Blood,n (%)** | | | | | | |
| At most 5.3% | 181(5.6) | 118(4.4) | 0.054 | 98(4.5) | 95(4.4) | 0.007 |
| 5.3-5.7% | 228(7) | 165(6.2) | 0.035 | 132(6.1) | 127(5.8) | 0.01 |
| 5.7-6.1% | 492(15.2) | 440(16.4) | 0.035 | 333(15.3) | 334(15.3) | 0.001 |
| 6.1-6.5% | 810(24.9) | 660(24.7) | 0.007 | 518(23.8) | 528(24.3) | 0.011 |
| 6.5-6.9% | 975(30) | 788(29.4) | 0.013 | 622(28.6) | 639(29.4) | 0.017 |
| 6.9-7.3% | 1060(32.6) | 807(30.1) | 0.054 | 668(30.7) | 682(31.3) | 0.014 |
| At least 7.3% | 2022(62.3) | 1394(52.1) | 0.207 | 1236(56.8) | 1242(57.1) | 0.006 |
| **Blood Pressure, Systolic, n (%)** | | | | | | |
| At most 120 mmHg | 1896(58.4) | 1556(58.1) | 0.005 | 1254(57.6) | 1259(57.8) | 0.005 |
| 120–130 mmHg | 2032(62.6) | 1668(62.3) | 0.006 | 1353(62.2) | 1342(61.6) | 0.01 |
| 130–140 mmHg | 2084(64.2) | 1716(64.1) | 0.002 | 1399(64.3) | 1389(63.8) | 0.01 |

*(Continued)*

**Table 1.** (Continued)

| Variables | Before the propensity score match | | | After the propensity score match | | |
|---|---|---|---|---|---|---|
| | GLP-1 RAs (n = 3,247) | SGLT2 inhibitors (n = 2,677) | SMD | GLP-1 RAs (n = 2,177) | SGLT2 inhibitors (n = 2,177) | SMD |
| 140–150 mmHg | 1950(60.1) | 1614(60.3) | 0.005 | 1310(60.2) | 1307(60) | 0.003 |
| 150–160 mmHg | 1701(52.4) | 1393(52) | 0.007 | 1140(52.4) | 1126(51.7) | 0.013 |
| At least 160 mmHg | 1562(48.1) | 1256(46.9) | 0.024 | 1030(47.3) | 1022(46.9) | 0.007 |
| **Blood pressure, diastolic, n (%)** | | | | | | |
| 0–60 mmHg | 1411(43.5) | 1163(43.4) | 0 | 945(43.4) | 926(42.5) | 0.018 |
| 60–70 mmHg | 1990(61.3) | 1665(62.2) | 0.019 | 1331(61.1) | 1328(61) | 0.003 |
| 70–80 mmHg | 2183(67.2) | 1809(67.6) | 0.007 | 1468(67.4) | 1456(66.9) | 0.012 |
| 80–90 mmHg | 2033(62.6) | 1667(62.3) | 0.007 | 1350(62) | 1357(62.3) | 0.007 |
| 90–100 mmHg | 1417(43.6) | 1077(40.2) | 0.069 | 907(41.7) | 900(41.3) | 0.007 |
| At least 100 mmHg | 847(26.1) | 603(22.5) | 0.083 | 530(24.3) | 525(24.1) | 0.005 |
| **BMI** | | | | | | |
| At most 18.5 kg/m2 | 65(2) | 40(1.5) | 0.039 | 47(2.2) | 37(1.7) | 0.033 |
| 18.5–25 kg/m2 | 83(2.6) | 144(5.4) | 0.145 | 72(3.3) | 69(3.2) | 0.008 |
| 25–30 kg/m2 | 238(7.3) | 382(14.3) | 0.225 | 203(9.3) | 211(9.7) | 0.013 |
| 30–35 kg/m2 | 484(14.9) | 517(19.3) | 0.117 | 368(16.9) | 372(17.1) | 0.005 |
| 35–40 kg/m2 | 578(17.8) | 439(16.4) | 0.037 | 394(18.1) | 385(17.7) | 0.011 |
| At least 40 kg/m2 | 664(20.5) | 419(15.7) | 0.125 | 398(18.3) | 390(17.9) | 0.01 |

The recording of data such as sex, race, HbA1c, blood pressure (systolic and diastolic), BMI, etc., may not be documented for every patient, hence the total sum may not reach 100%. GLP-1 RAs, glucagon-like peptide-1 receptor agonists; SGLT2 inhibitors, Sodium-glucose cotransporter-2 inhibitors; SMD, standardized mean difference.

The EMPEROR-Preserved and DELIVER trials demonstrated the efficacy of SGLT2 inhibitors in reducing heart failure hospitalizations and improving outcomes in HFpEF patients. While these studies solidify the role of SGLT2 inhibitors, our research addresses a critical gap by directly comparing these agents with GLP-1 receptor agonists, which offer complementary benefits such as anti-inflammatory and metabolic effects. This comparison provides novel insights into tailoring therapies for HFpEF diabetic patients [5,10]. The PRESERVED-HF trial with Dapagliflozin showed promising results with a 5.8-point increase in KCCQ-CSS at 12 weeks [11]. The STEP-HFpEF trial with GLP-1 RAs semaglutide on HFpEF and obese patients demonstrated enhanced exercise capacity (7.9-point increase in KCCQ-CSS compared to placebo over 52 weeks) but lacked sufficient evidence on cardiovascular endpoints [9,11,12]. In these selected clinical trials, GLP-1 RAs have demonstrated notable improvement in clinical symptoms among obese HFpEF patients, primarily associated with weight reduction. Conversely, SGLT2 inhibitors showed lower primary cardiovascular endpoint rates with or without diabetes mellitus and have exhibited a quicker onset of symptom alleviation.

Although HFpEF and HFrEF are distinct diseases, evidence suggests potential transitions from HFpEF to HFrEF [4]. GLP-1 receptor agonists (GLP-1 RAs) have demonstrated significant reductions in major adverse cardiovascular events and cardiometabolic benefits, prompting investigations into their role in HFpEF management. Preclinical studies have shown that liraglutide and semaglutide improve body weight, cardiac function, and structural parameters in obesity-related HFpEF models, with benefits extending beyond weight loss. The STEP-HFpEF trial further supports semaglutide potential to prevent HFpEF progression by reducing pathological ventricular remodeling through significant weight reduction [9,13]. Ongoing clinical trials, such as SUMMIT (NCT04847557), aim to validate these preclinical findings. If successful, GLP-1

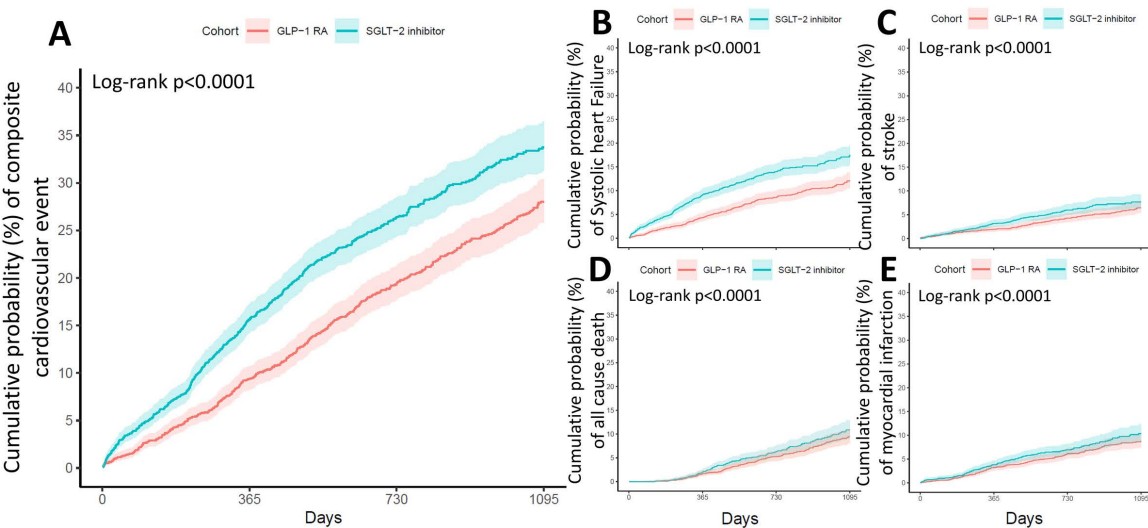

| Patients with outcome (n=2177 patients per cohort after PSM) | | | |
|---|---|---|---|
| Outcome | GLP-1 RA | SGLT-2 inhibitor | Hazard ratio (95% CI) |
| **Composite cardiovascular endpoint** | | | |
| 1 year after the index date | 201 | 334 | 0.577 (0.484–0.687) |
| 2 years after the index date | 368 | 497 | 0.667 (0.583–0.763) |
| 3 years after the index date | 466 | 555 | 0.713 (0.630–0.807) |
| **All-cause mortality** | | | |
| 1 year after the index date | 35 | 41 | 0.840 (0.535–1.318) |
| 2 years after the index date | 94 | 106 | 0.803 (0.608–1.060) |
| 3 years after the index date | 141 | 138 | 0.834 (0.659–1.056) |
| **Systolic heart failure** | | | |
| 1 year after the index date | 93 | 195 | 0.461 (0.360–0.591) |
| 2 years after the index date | 163 | 265 | 0.566 (0.466–0.689) |
| 3 years after the index date | 200 | 289 | 0.605 (0.505–0.725) |
| **Acute myocardial infarction** | | | |
| 1 year after the index date | 69 | 81 | 0.842 (0.611–1.161) |
| 2 years after the index date | 115 | 127 | 0.846 (0.657–1.089) |
| 3 years after the index date | 143 | 151 | 0.825 (0.656–1.038) |
| **Stroke** | | | |
| 1 year after the index date | 42 | 67 | 0.621 (0.422–0.913) |
| 2 years after the index date | 79 | 105 | 0.693 (0.518–0.929) |
| 3 years after the index date | 102 | 117 | 0.752 (0.576–0.982) |

**Fig 2. Effects of GLP-1 RAs and SGLT2 Inhibitors on Composite Cardiovascular Endpoint.** This figure illustrates the effects of GLP-1 receptor agonists (GLP-1 RAs) and SGLT2 inhibitors on composite cardiovascular outcomes, including all-cause mortality, progression to systolic heart failure, acute myocardial infarction, and stroke, after propensity score matching. GLP-1 RAs, glucagon-like peptide-1 receptor agonists; SGLT2 inhibitors, sodium-glucose cotransporter-2 inhibitors.

**Fig 3. Kaplan-Meier Estimates of Primary and Secondary Outcomes in Patients with DM and HFpEF Treated with GLP-1 RAs or SGLT2 Inhibitors.** Kaplan-Meier survival curves compare the two treatment groups across primary and secondary outcomes, including (A) composite cardiovascular events, (B) systolic heart failure (SHF), (C) stroke, (D) all-cause mortality, and (E) acute myocardial infarction (MI). DM, diabetes mellitus; HFpEF, heart failure with preserved ejection fraction; GLP-1 RAs, glucagon-like peptide-1 receptor agonists; SGLT2 inhibitors, sodium-glucose cotransporter-2 inhibitors.

| Patients with composite cardiovascular event | | | |
|---|---|---|---|
| Subgroup | GLP-1 RA | SGLT-2 inhibitor | Hazard ratio (95% CI) |
| Overall ( 2177 patients per cohort) | 466 | 555 | 0.713 (0.630–0.807) |
| Age | | | |
| >=60 (1405 patients per cohort) | 87 | 117 | 0.658 (0.498–0.869) |
| <60 (510 patients per cohort) | 323 | 371 | 0.723 (0.622–0.840) |
| Gender | | | |
| Male (964 patients per cohort) | 222 | 265 | 0.724 (0.605–0.866) |
| Female (784 patients per cohort) | 158 | 200 | 0.609 (0.493–0.751) |
| Races | | | |
| White (1194 patients per cohort) | 266 | 315 | 0.728 (0.618–0.857) |
| Black (428 patients per cohort) | 98 | 104 | 0.788 (0.597–1.039) |
| Others (323 patients per cohort) | 75 | 81 | 0.801 (0.585–1.098) |
| Moderate to severe CKD | | | |
| Yes (887 patients per cohort) | 251 | 273 | 0.745 (0.627–0.885) |
| No (1069 patients per cohort) | 180 | 238 | 0.685 (0.564–0.831) |
| Chronic ischemic heart disease | | | |
| Yes (1044 patients per cohort) | 265 | 288 | 0.779 (0.659–0.921) |
| No (955 patients per cohort) | 177 | 221 | 0.664 (0.543–0.811) |
| HbA1c | | | |
| >=8.0% (486 patients per cohort) | 134 | 142 | 0.803 (0.634–1.017) |
| <8.0% (1495 patients per cohort) | 314 | 370 | 0.727 (0.625–0.846) |

Adjusted hazard ratio: 0.4    0.7    1    1.5

**Fig 4. Subgroup Analysis of Composite Cardiovascular Endpoint Risk.** This figure presents the subgroup analysis of the risk for the composite cardiovascular endpoint, stratified by age, gender, race, moderate-to-severe chronic kidney disease (CKD), chronic ischemic heart disease, and diabetes mellitus (DM) control, after propensity score matching. CKD, chronic kidney disease; DM, diabetes mellitus.

RAs like semaglutide could transform HFpEF management by addressing weight-related and cardiometabolic issues, preventing progression to HFrEF.

This study demonstrated that GLP-1 RAs reduced the risk of stroke in patients with DM and HFpEF. The REWIND and SUSTAIN-6 trials consistently highlighted the cardiovascular benefits of GLP-1 RAs, particularly in reducing stroke risk through their anti-inflammatory and atherosclerosis-reducing properties [12,14]. Comparatively, a cohort study found that SGLT2 inhibitors were associated with a 20% reduction in stroke incidence in patients with DM and atrial fibrillation [15]. However, meta-analyses suggest SGLT2 inhibitors may have limited benefits for ischemic stroke compared to GLP-1 RAs [16]. At the molecular level, GLP-1 RAs provide neuroprotective effects by mitigating oxidative damage and apoptosis, promoting neuronal growth, and preventing amyloid plaque deposition [17]. By preserving neuronal health and reducing atherosclerotic changes, GLP-1 RAs effectively lower stroke risk in patients with DM and HFpEF.

In this study, GLP-1 RAs demonstrated a protective trend against acute myocardial infarction (MI) compared to SGLT2 inhibitors, though the results were not statistically significant. A systematic review by Giugliano et al., encompassing eight cardiovascular safety trials with over 60,000 patients, reported a protective effect of GLP-1 RAs on fatal and non-fatal MI (HR 0.90, [95% CI, 0.83–0.98]; $p = 0.020$) [18].

In contrast, a population-based study evaluating SGLT2 inhibitors in patients with type 2 DM undergoing PCI for MI found no significant effect on non-fatal MI prevention (HR 0.98, [95% CI, 0.68–1.40]) [19]. These findings highlight the need for further research to clarify the role of GLP-1 RAs in acute MI prevention, particularly in patients with HFpEF. The observed trend suggests potential benefits that warrant additional investigation.

This study found that GLP-1 RAs demonstrated greater efficacy in individuals of White ethnicity than SGLT2 inhibitors, while no significant differences were observed among Black or other racial groups. While clinical trials like LEADER and SUSTAIN have shown consistent cardiovascular benefits across diverse populations, disparities in access to GLP-1 RAs among Asian, Black, and Hispanic individuals have been noted [12,20]. Further research is needed to understand these

ethnic variations better. Additionally, no significant differences in cardiovascular protection were found between the two therapies in patients with poorly controlled diabetes (HbA1c ≥ 8.0). Both SGLT2 inhibitors and GLP-1 RAs have shown cardiovascular benefits in such populations. For example, Ceriello et al. demonstrated that early administration of SGLT2 inhibitors significantly mitigated the association between poorly controlled diabetes (HbA1c > 7%) and adverse cardiovascular outcomes within two years of diagnosis [21]. These findings highlight the importance of equitable access and early intervention for optimizing treatment benefits. The subgroup analysis highlights that GLP-1 receptor agonists may provide more significant benefits in older adults, women, and patients with better glycemic control. These findings underscore the importance of individualized treatment approaches in managing HFpEF with diabetes. Clinicians may prioritize GLP-1 RAs for patients who align with these characteristics to maximize therapeutic outcomes. Future research should explore the mechanisms underlying these subgroup differences and evaluate the efficacy of combining GLP-1 RAs with SGLT2 inhibitors to leverage their complementary benefits in diverse patient populations.

While this propensity score-matched study presents significant findings, it is important to acknowledge some limitations that warrant further consideration. The reliance on database-driven diagnoses may affect the precision of HFpEF identification due to the absence of key clinical data, such as NT-proBNP, NYHA functional class, and ejection fraction. While the ICD-10 code for diastolic HF (I50.3) has a positive predictive value of 92% for HFpEF cases with EF > 50%, the lack of detailed clinical information may introduce bias, as patients with more severe HFpEF might be preferentially treated with SGLT2 inhibitors. The high prevalence of obesity (BMI > 30 kg/m²) among both cohorts could influence outcomes, potentially favoring GLP-1 RAs. Additionally, the database did not provide details on hospitalization etiology, initial medication dosages, or administration routes, limiting the scope of comparative analysis. The inability to distinguish between fatal and non-fatal MI and strokes further constrained the findings. Finally, as a retrospective study, unmeasured variables may have introduced residual confounding despite rigorous matching techniques. These limitations highlight the need for prospective studies to confirm and expand upon these findings.

This retrospective cohort study found GLP-1 RAs were more effective than SGLT2 inhibitors in reducing the 3-year risk of composite cardiovascular events, especially SHF progression and stroke in patients with HFpEF and DM. More significant benefits were observed in subgroups aged ≥60, women, Caucasians, those without moderate-to-severe chronic kidney disease or chronic ischemic heart disease, and those with controlled DM. Further randomized trials are needed to confirm these findings and clarify this high-risk population's optimal cardiovascular protection therapy.

## Supporting information

**S1 File. Codes run in TriNetX database.**
(DOCX)

**S2 File. Values for Kaplan-Meier Estimates of Primary and Secondary Outcomes in Patients with DM and HFpEF Treated with GLP-1 RAs or SGLT2 Inhibitors.**
(XLSX)

## Acknowledgments

The authors wish to acknowledge TriNetx for providing the statistical database, Chung Shan Medical University for statistical assistance, and Shin Kong Wu Ho-Su Memorial Hospital for their assistance in preparing the manuscript.

## Author contributions

**Conceptualization:** Allen Cheng-Wei Li.

**Data curation:** Yang-Chi Lin, Lung-Ching Chen.

**Formal analysis:** Lung-Ching Chen.

**Investigation:** Jing-Yang Huang.

**Methodology:** Allen Cheng-Wei Li, Yang-Chi Lin, Jing-Yang Huang.

**Software:** Yang-Chi Lin.

**Supervision:** Su-Kiat Chua.

**Validation:** Su-Kiat Chua.

**Writing – original draft:** Allen Cheng-Wei Li, Yang-Chi Lin.

**Writing – review & editing:** Su-Kiat Chua.

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
