## [Decision Letter · Decision Letter 0]

PONE-D-24-35455Superior Cardiovascular Protection with GLP-1 RAs Over SGLT2 Inhibitors in DM and HFpEF: A Propensity Score Matching StudyPLOS ONE

Dear Dr. Chua,

Thank you for submitting your manuscript to PLOS ONE. After careful consideration, we feel that it has merit but does not fully meet PLOS ONE’s publication criteria as it currently stands. Therefore, we invite you to submit a revised version of the manuscript that addresses the points raised during the review process.

We look forward to receiving your revised manuscript.

Kind regards,

Ahmed Mustafa Rashid

Academic Editor

PLOS ONE

Journal Requirements:

Please confirm at this time whether or not your submission contains all raw data required to replicate the results of your study. Authors must share the “minimal data set” for their submission. PLOS defines the minimal data set to consist of the data required to replicate all study findings reported in the article, as well as related metadata and methods (https://journals.plos.org/plosone/s/data-availability#loc-minimal-data-set-definition ).

If your submission does not contain these data, please either upload them as Supporting Information files or deposit them to a stable, public repository and provide us with the relevant URLs, DOIs, or accession numbers. For a list of recommended repositories, please see https://journals.plos.org/plosone/s/recommended-repositories .

3. If any table files for review show as item type ‘other’ please change to item type ‘table’ as the reviewer does not have access to these ’other’ files.

Reviewers' comments:

Reviewer's Responses to Questions

**Comments to the Author**

1. Is the manuscript technically sound, and do the data support the conclusions?

Reviewer #1: Yes

Reviewer #2: Yes

Reviewer #3: Yes

2. Has the statistical analysis been performed appropriately and rigorously? 

Reviewer #1: Yes

Reviewer #2: I Don't Know

Reviewer #3: Yes

3. Have the authors made all data underlying the findings in their manuscript fully available?

Reviewer #1: Yes

Reviewer #2: Yes

Reviewer #3: Yes

4. Is the manuscript presented in an intelligible fashion and written in standard English?

Reviewer #1: Yes

Reviewer #2: Yes

Reviewer #3: Yes

5. Review Comments to the Author

Reviewer #1: This propensity score match study by Li et al. provides valuable insights into the cardiovascular protective effects of GLP-1 receptor agonists compared to SGLT2 inhibitors in patients with HFpEF and diabetes. Further comments are outlined.

Reviewer #2: Page 11, Line 88: The sentence “affecting nearly 25% of diagnosed HFpEF cases” could benefit from clarification. It’s not immediately clear whether this percentage refers to the overall population of HFpEF patients or a specific subgroup.

Page 11, Lines 89-92: The list of comorbidities is informative but might feel overwhelming. You could group them or highlight the most significant ones related to your study to enhance clarity.

Page 11, Lines 101-104: When mentioning the EMPEROR-Preserved trial, clarify the significance of the 29% reduction in hospitalization. Why is this finding crucial for your study?

Page 13, Lines 141-145: The inclusion criteria for the cohorts are complex. Consider simplifying the language or breaking this into shorter sentences to enhance clarity, especially the definition of "long-term use."

Page 13, Lines 162-164: While you mention that flowcharts are presented in Figure 1, a brief description of what these flowcharts illustrate would be helpful.

Page 14, Lines 165-167: The mention of numerical laboratory data is important. However, a brief explanation of how these specific measures relate to HF progression would provide additional context.

Page 15, Lines 193-199: The statistical analysis section is thorough, but consider mentioning the software or tools used for the analysis, as this information is often valuable for reproducibility.

Page 18, Lines 275-277: While referencing the EMPEROR PRESERVED and DELIVER trials adds credibility, it would strengthen your argument to clarify how these studies directly relate to the findings of your research on GLP-1 RAs.

Page 20, Lines 321-324: The transition between the evidence from REWIND and SUSTAIN-6 could be more fluid. Consider linking these studies by discussing common themes in their findings.

Page 21, Lines 336-339: You outline the protective effects of GLP-1 RAs well. Strengthening this section with comparisons to SGLT2 inhibitors could clarify how these treatments differ in their mechanisms.

Page 22, Lines 368-375: The conclusion of the subgroup analysis could be more impactful by explicitly stating what these findings mean for clinical practice and suggesting areas for future research.

Page 30-33: Consider replacing the blurry images with high-definition (HD) images. HD images improve clarity and detail, making it easier for readers to interpret the data. Clear visuals enhance the professionalism of the paper and aid in effective communication of key findings, ultimately improving reader engagement and understanding.

Reviewer #3: The manuscript is well-written and presents the data clearly. Only minor proofreading is needed to address a few grammatical and typographical errors. Overall, the paper is strong and ready for publication with minimal revisions.

6. PLOS authors have the option to publish the peer review history of their article (what does this mean? ). If published, this will include your full peer review and any attached files.

**Do you want your identity to be public for this peer review?** For information about this choice, including consent withdrawal, please see our Privacy Policy .

Reviewer #1: No

Reviewer #2: **Yes: ** Akash Kumar

Reviewer #3: No

---

## [Author Response · Author response to Decision Letter 1]

22 Jan 2025

Response to Reviewer 1

Line 89-98: Overall, the introduction flows well, but it lacks some key details regarding HFpEF and DM, which weakens the rationale for the study. To strengthen the introduction, I recommend including more specific information about the pathophysiology of HFpEF in the context of diabetes and expanding on why a direct comparison of GLP-1 receptor agonists and SGLT2 inhibitors is crucial for this population.

Response: Thank you for your valuable suggestion. We have expanded the introduction to include specific details about the pathophysiology of HFpEF in the context of diabetes and to emphasize the importance of comparing GLP-1 receptor agonists and SGLT2 inhibitors in this population. This addition highlights the mechanisms linking HFpEF and DM, such as insulin resistance, inflammation, myocardial fibrosis, and endothelial dysfunction, contributing to poor cardiovascular outcomes. (Lines 93-104 of the revised article)

The discussion is well written; however, the use of headings disrupts the flow and cohesiveness of the text. I recommend removing these headings and rephrasing to enhance continuity between paragraphs, which would create a smoother reading experience.

Response: Thank you for your suggestion regarding the use of headings in the discussion section. We have removed the subheadings in the discussion section and rephrased transitions to create a more cohesive and natural flow. This restructuring ensures the narrative reads smoothly without abrupt breaks.

Line 263-267: “ This study found that GLP-1 RAs reduced the risk of cardiovascular events, progression to SHF, and stroke compared with SGLT2 inhibitors in patients with HFpEF and DM. The benefits appeared to diminish slightly over the 3-year follow-up. GLP-1 RAs also showed a trend toward reduced MI and mortality, although this was not statistically significant “ should be rephrased to: Our results demonstrate that GLP-1 receptor agonists provide better cardiovascular protection than SGLT2 inhibitors in patients with HFpEF and DM, significantly reducing the risk of cardiovascular events, progression to SHF, and stroke. While the benefits were strongest in the initial treatment stages, there was a slight reduction in efficacy over the 3-year follow-up. Additionally, GLP-1 RAs showed a trend towards fewer MI’s and lower mortality rates, though these findings were not statistically significant.

Response: We revised this section following your suggestion to improve readability and emphasize key findings. (Lines 255-261 of the revised article)

Line 270-271: This point has been reiterated in lines 286-288. Since the rationale is already outlined in the introduction (lines 117-121), it is unnecessary to restate it here.

Response: Thank you for pointing out the redundancy. We have removed the redundant statement to streamline the discussion and avoid repetition, as the rationale is already provided in the introduction.

Line 378: “ However, some limitations of this study warrant further consideration “. This starting sentence seems incomplete, it should be rephrased to: While this propensity score-matched study presents significant findings, it is important to acknowledge some limitations that warrant further consideration.

Response: We revised the opening of the limitations section to make it complete and more balanced. (Line 337-338 of the revised article)

Response to Reviewer 2:

Reviewer #2: Page 11, Line 88: The sentence “affecting nearly 25% of diagnosed HFpEF cases” could benefit from clarification. It’s not immediately clear whether this percentage refers to the overall population of HFpEF patients or a specific subgroup.

Response: We have clarified this sentence to specify that the percentage refers to the proportion of HFpEF patients diagnosed with diabetes mellitus (DM). (Lines 88-90 of the revised article)

Page 11, Lines 89-92: The list of comorbidities is informative but might feel overwhelming. You could group them or highlight the most significant ones related to your study to enhance clarity.

Response: We grouped the comorbidities into broader categories and highlighted those most relevant to the study. (Lines 90-93 of the revised article)

Page 11, Lines 101-104: When mentioning the EMPEROR-Preserved trial, clarify the significance of the 29% reduction in hospitalization. Why is this finding crucial for your study?

Response: Thank you for your insightful comment regarding the EMPEROR-Preserved trial and the significance of the 29% reduction in hospitalization. We understand the importance of clarifying how this finding is relevant to our study. To address your suggestion, we have revised the text to explain why this result is a critical reference point for evaluating the comparative efficacy of GLP-1 receptor agonists and SGLT2 inhibitors in HFpEF patients with diabetes. This context strengthens the rationale for our direct comparison study. (Lines 107-110 of the revised article)

Page 13, Lines 141-145: The inclusion criteria for the cohorts are complex. Consider simplifying the language or breaking this into shorter sentences to enhance clarity, especially the definition of "long-term use."

Response: Thank you for your helpful suggestion regarding the complexity of the inclusion criteria and the definition of "long-term use." We have revised this section to ensure that the inclusion criteria are more concise and easier to understand, with a more precise definition of "long-term use." (Lines 141-144 of the revised article)

Page 13, Lines 162-164: While you mention that flowcharts are presented in Figure 1, a brief description of what these flowcharts illustrate would be helpful.

Response: Thank you for pointing out the need for a brief description of what the flowcharts in Figure 1 illustrate. We agree that adding this information will enhance the figure's clarity and utility for readers. In response, we have revised the text to include a concise explanation of the flowcharts' content and purpose. (Lines 149-153 of the revised article)

Page 14, Lines 165-167: The mention of numerical laboratory data is important. However, a brief explanation of how these specific measures relate to HF progression would provide additional context.

Response: Thank you for highlighting the importance of providing additional context for the numerical laboratory data and their relationship to HF progression. We agree that this explanation will enhance the readers' understanding of the relevance of these measures. We have revised the text to briefly explain how these specific variables are associated with HF progression. (Lines 164-169 of the revised article)

Page 15, Lines 193-199: The statistical analysis section is thorough, but consider mentioning the software or tools used for the analysis, as this information is often valuable for reproducibility.

Response: Thank you for suggesting that we include the software and tools used for the analysis. We have revised the text to specify the platforms and software used for data analysis and figure generation. (Lines 197-201 of the revised article)

Page 18, Lines 275-277: While referencing the EMPEROR PRESERVED and DELIVER trials adds credibility, it would strengthen your argument to clarify how these studies directly relate to the findings of your research on GLP-1 RAs.

Response: Thank you for pointing out the need to clarify the connection between the EMPEROR-Preserved and DELIVER trials and our findings on GLP-1 receptor agonists (GLP-1 RAs). We have revised the text to highlight how these studies provide a comparative foundation and underscore the unique contributions of GLP-1 RAs to HFpEF management. (Lines 262-268 of the revised article)

Page 20, Lines 321-324: The transition between the evidence from REWIND and SUSTAIN-6 could be more fluid. Consider linking these studies by discussing common themes in their findings.

Response: Thank you for your insightful suggestion to improve the transition between the REWIND and SUSTAIN-6 studies. We agree that linking these studies through common themes in their findings will create a smoother flow and provide a more cohesive narrative. We have revised the text to highlight shared outcomes and mechanisms observed in both studies. (Lines 292-294 of the revised article)

Page 21, Lines 336-339: You outline the protective effects of GLP-1 RAs well. Strengthening this section with comparisons to SGLT2 inhibitors could clarify how these treatments differ in their mechanisms.

Response: Thank you for your suggestion to strengthen the discussion on the protective effects of GLP-1 receptor agonists (GLP-1 RAs) by comparing them with SGLT2 inhibitors. We have revised the text to include a direct comparison, emphasizing how the complementary mechanisms of these therapies impact cardiovascular outcomes. (Lines 292-302 of the revised article)

Page 22, Lines 368-375: The conclusion of the subgroup analysis could be more impactful by explicitly stating what these findings mean for clinical practice and suggesting areas for future research.

Response: Thank you for your thoughtful suggestion to enhance the conclusion of the subgroup analysis by explicitly addressing its implications for clinical practice and suggesting areas for future research. We have revised the text to clearly state how these findings can guide clinical decision-making and highlight opportunities for further investigation. (Lines 327-335 of the revised article)

Page 30-33: Consider replacing the blurry images with high-definition (HD) images. HD images improve clarity and detail, making it easier for readers to interpret the data. Clear visuals enhance the professionalism of the paper and aid in effective communication of key findings, ultimately improving reader engagement and understanding.

Response: Thank you for suggesting replacing the blurry images with high-definition (HD) versions. We have responded by replacing all relevant figures with HD images to ensure clarity and a better presentation of our key findings.

Response to reviewer 3:

Reviewer #3: The manuscript is well-written and presents the data clearly. Only minor proofreading is needed to address a few grammatical and typographical errors. Overall, the paper is strong and ready for publication with minimal revisions.

Response: Thank you very much for your positive feedback and for acknowledging the clarity and quality of our manuscript. We appreciate your time and effort in reviewing our work.

As you recommended, we have carefully reviewed the manuscript for grammatical and typographical errors and made the necessary corrections to ensure the text is clear and polished.

---

## [Decision Letter · Decision Letter 1]

Superior Cardiovascular Protection with GLP-1 RAs Over SGLT2 Inhibitors in DM and HFpEF: A Propensity Score Matching Study

PONE-D-24-35455R1

Dear Dr. Chua,

We’re pleased to inform you that your manuscript has been judged scientifically suitable for publication and will be formally accepted for publication once it meets all outstanding technical requirements.

Kind regards,

Abdelrahman M. Attia

Academic Editor

PLOS ONE

---

## [Editor Report · Acceptance letter]

PONE-D-24-35455R1

PLOS ONE

Dear Dr. Chua,

I'm pleased to inform you that your manuscript has been deemed suitable for publication in PLOS ONE. Congratulations! Your manuscript is now being handed over to our production team.

Kind regards,

on behalf of

Dr. PLOS Manuscript Reassignment

Staff Editor

PLOS ONE